# Electrochemical Oxidation of an Organic Dye Adsorbed on Tin Oxide and Antimony Doped Tin Oxide Graphene Composites

**Farbod Sharif** and **Edward P. L. Roberts** *

University of Calgary, Department of Chemical and Petroleum Engineering, 2500 University Drive NW, Calgary, AB T2N 1N4, Canada; shariff@ucalgary.ca
* Correspondence: edward.roberts@ucalgary.ca; Tel.: +1-403-220-4466

**Abstract:** Electrochemical regeneration suffers from low regeneration efficiency due to side reactions like oxygen evolution, as well as oxidation of the adsorbent. In this study, electrically conducting nanocomposites, including graphene/$SnO_2$ (G/$SnO_2$) and graphene/Sb-$SnO_2$ (G/Sb-$SnO_2$) were successfully synthesized and characterized using nitrogen adsorption, scanning electron microscopy, transmission electron microscopy, and Raman spectroscopy. Thereafter, the adsorption and electrochemical regeneration performance of the nanocomposites were tested using methylene blue as a model contaminant. Compared to bare graphene, the adsorption capacity of the new composites was ≥40% higher, with similar isotherm behavior. The adsorption capacity of G/$SnO_2$ and G/Sb-$SnO_2$ were effectively regenerated in both NaCl and $Na_2SO_4$ electrolytes, requiring as little charge as 21 C mg$^{-1}$ of adsorbate for complete regeneration, compared to >35 C mg$^{-1}$ for bare graphene. Consecutive loading and anodic electrochemical regeneration cycles of the nanocomposites were carried out in both NaCl and $Na_2SO_4$ electrolytes without loss of the nanocomposite, attaining high regeneration efficiencies (ca. 100%).

**Keywords:** graphene; tin oxide; antimony doped tin oxide; adsorption; electrochemical oxidation

## 1. Introduction

Adsorption is a promising method for the removal of soluble and insoluble organics from wastewater effluents due to ease of operation, a wide range of applications, and a high level of purity of the treated water [1]. However, handling of the loaded adsorbent is a challenge because of factors that constrain disposal, including the toxicity of the adsorbate and the high cost of replacement adsorbent [2,3]. Electrochemical regeneration has shown potential for effectively recovering the adsorptive capacity of the loaded adsorbents [4–7].

Early studies of electrochemical regeneration with an activated carbon adsorbent showed that although it has a high adsorptive capacity, it requires long regeneration times, resulting in high energy consumption. This is mainly due to the porous surface and low conductivity of the activated carbon [8–10]. Brown et al. [11–13] studied an alternative material, graphite intercalation compound (GIC), which is nonporous and has high electrical conductivity. This material demonstrated high regeneration efficiency but low adsorptive capacity. In the previous study [14], reduced graphene oxide (rGO)/magnetite was chosen as an adsorbent since it has a nonporous surface, high surface area, and high electrical conductivity. The findings revealed that rGO/magnetite has satisfactory adsorptive capacity which can be completely regenerated. However, the electrochemical regeneration process caused oxidation and corrosion of the adsorbent. Similar adsorbent oxidation was also seen by Nkrumah-Amoako et al. [15] for a GIC adsorbent. Graphitic materials display behaviors of both active

and inactive electrodes, but the dominant mechanism for organic oxidation is believed to be direct electron transfer [16]. This is likely the main reason for oxidation of the graphene as well.

It is essential to tackle the problem of adsorbent oxidation, and this can be done by changing the dominant mechanism from direct electron transfer to mediated degradation using electro-generated hydroxyl radicals. One approach would be to coat the surface of the carbon adsorbent, i.e., graphene, with materials that behave as inactive anodes, such as boron-doped diamond (BDD), $SnO_2$, $Sb$-$SnO_2$, or $PbO_2$. Due to the high overpotential of the inactive materials for oxygen evolution, they are considered good candidates for organic oxidation by means of reactive oxygen species such as hydroxyl radicals. Extensive research has been conducted on inactive materials to increase the rate of reactive oxygen species production. Ozone production on $Sb$-$SnO_2$/Ti, hydroxyl radical generation on fluoride-doped lead oxide, hydrogen peroxide generation on niobium lead oxide, and hydroxyl radical generation on BDD and $TiO_2$/Ti are several good examples [17–23]. These materials increase the oxygen evolution overpotential, minimizing this side reaction, thus increasing the current efficiency for organic oxidation [21,24,25]. Owing to its large band gap of 5.45 eV, BDD is the best inactive, corrosion-resistant material [26]; however, due to its high cost, BDD cannot be effectively used for adsorption and electrochemical regeneration. $PbO_2$ also has a high overpotential for oxygen evolution [27] as well as a much lower cost than BDD, but leaching of lead from the $PbO_2$ into the solution can contaminate the treated water [28].

In the current study, graphene/$SnO_2$ (G/$SnO_2$) and graphene/$Sb$-$SnO_2$ (G/$Sb$-$SnO_2$) nanocomposites were prepared, characterized, and utilized in an adsorption and electrochemical regeneration process. Our previous study on electrochemical regeneration of graphene $TiO_2$ adsorbents materials revealed that the addition of the $TiO_2$ nanoparticles to the surface of the graphene increases catalytic activity, reducing the required time for complete regeneration, and also protects the graphene surface from corrosion [29]. However, the prepared adsorbents did not demonstrate good adsorptive capacity due to loss of amorphous carbon during the calcination, required for synthesis of the $TiO_2$ nanocomposite.

The goal of the present study was to prepare new materials with higher adsorptive capacity and comparable catalytic activity to the graphene/$TiO_2$ nanocomposite which can be applied in successive adsorption electrochemical regeneration process. In addition to the methylene blue (MB) adsorption and electrochemical regeneration behavior of these adsorbents, their electrochemical properties were also investigated. The effect of the nanomaterial loading on the nanocomposites and the electrolyte types on the electrochemical regeneration and the durability of the new materials was also assessed.

## 2. Result and Discussion

### 2.1. Raman Spectra

The surface chemical composition of the graphene and composite samples was characterized by Raman spectroscopy, as shown in Figure 1. The peaks observed at 1358 and 1580 $cm^{-1}$ can be attributed to the D band and G band of the graphene, respectively [30]. The peak at a Raman shift of 572 $cm^{-1}$ can be assigned to the cassiterite $SnO_2$ nanoparticles; this peak is only present for nanometer-scale $SnO_2$ particles [31,32]. The four broad Raman peaks at 458, 566, 720, and 632 $cm^{-1}$ can be attributed to $SnO_2$ nanoparticles doped with antimony. The first three peaks are the surface vibration modes [33], and the last one is the vibration in the plane perpendicular to the c-axis [34]. Thus, the Raman spectra confirm the presence of $SnO_2$ and $Sb$-$SnO_2$ on the nanocomposite materials.

### 2.2. SEM and TEM

The morphology of the G/$SnO_2$ and G/$Sb$-$SnO_2$ nanocomposites were observed using TEM and SEM (Figures 2 and 3). A comparison between the SEM image of bare graphene, G/$SnO_2$, and G/$Sb$-$SnO_2$ reveals that $SnO_2$ nanoparticles have adhered to the surface of the graphene; however, due to the SEM limitations, it is not possible to make more quantitative comments on the nanoparticle distribution. The TEM images, Figure 3b,d show that the average sizes of $SnO_2$ and $Sb$-$SnO_2$ nanoparticles are around

5 and 30 nm, respectively. It can also be inferred that the nanoparticles have not been distributed uniformly on the surface; they tend to aggregate due to their high surface energy [35].

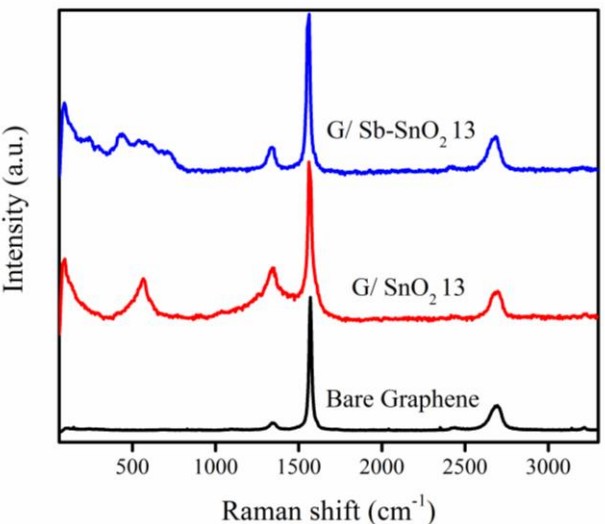

**Figure 1.** Raman spectra of Bare graphene, $SnO_2$, $Sb-SnO_2$, $G/SnO_2$, and $G/Sb-SnO_2$ nanocomposites.

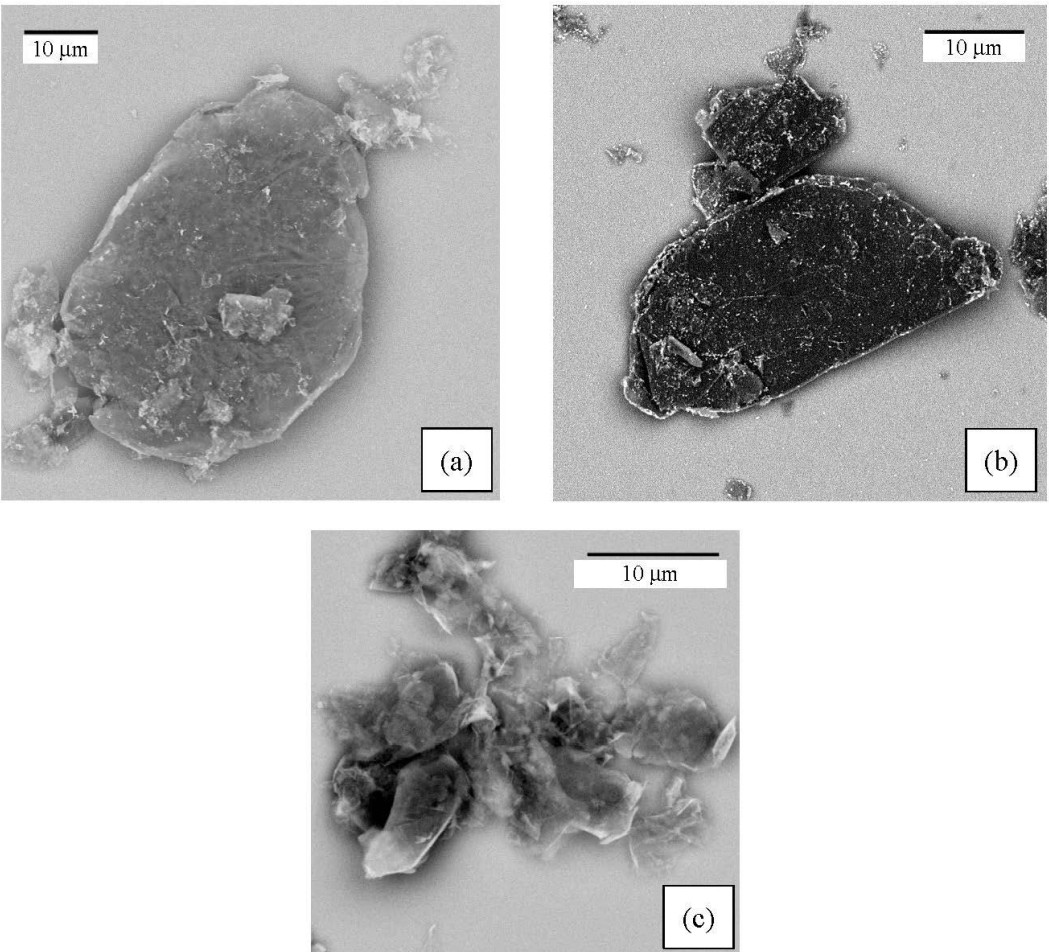

**Figure 2.** SEM images of the (**a**) $G/SnO_2$ nanocomposite, (**b**) $G/Sb-SnO_2$ nanocomposite, and (**c**) bare graphene.

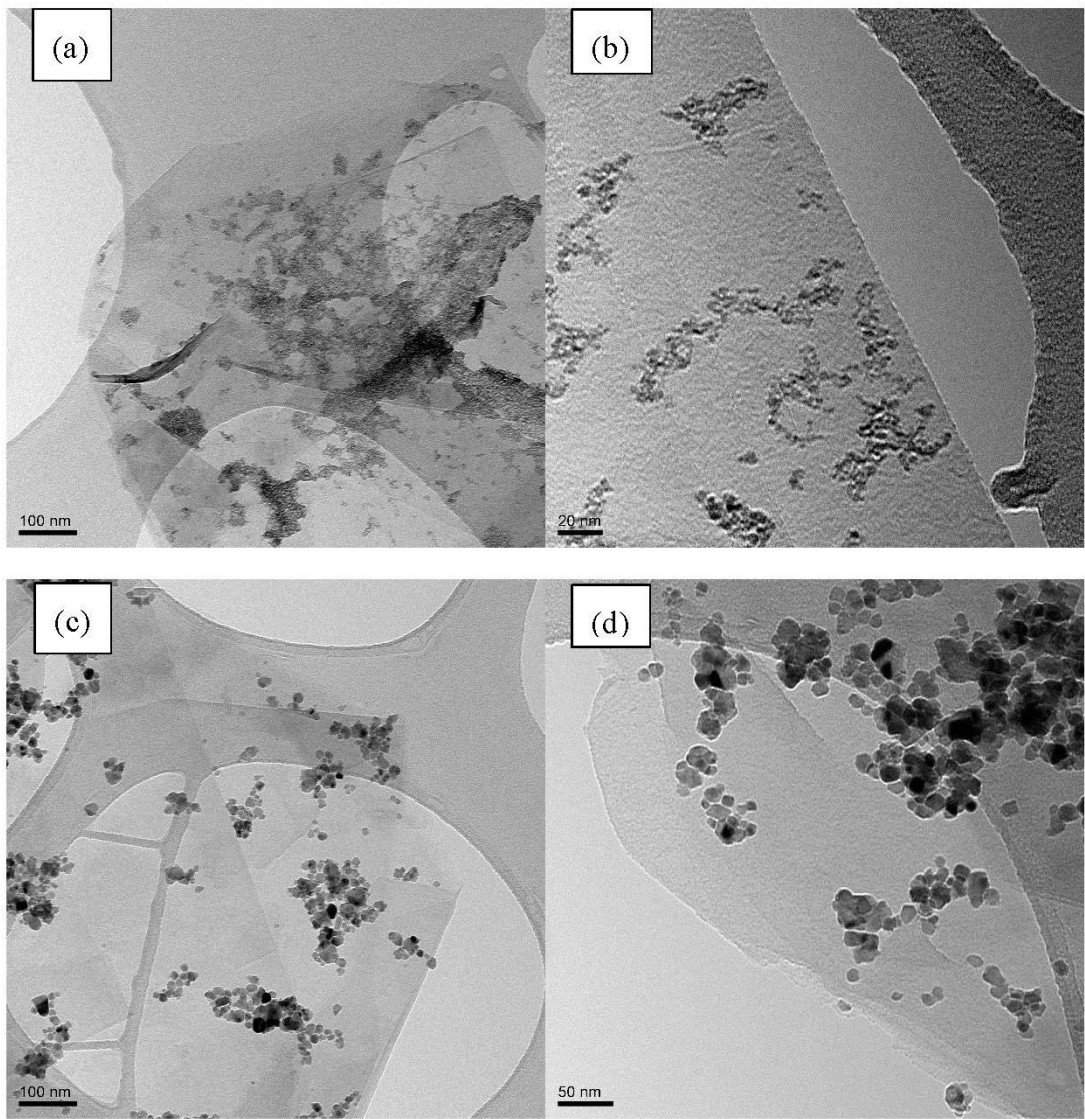

**Figure 3.** (**a**) Low magnification TEM image of the G/SnO2 nanocomposite (**b**) High-magnification TEM image of the G/SnO2 nanocomposite. (**c**) Low-magnification TEM image of the G/Sb-SnO2 nanocomposite. (**d**) High-magnification TEM image of the G/Sb-SnO2 nanocomposite.

### 2.3. Adsorption Study

Adsorption isotherm studies were carried out to find the adsorptive capacity of the prepared samples and to determine the adsorption range to be used for adsorbent regeneration studies. The results obtained are depicted in Figure 4. Table 1 shows the calculated surface area for each sample using the BET method. It can be seen that $G/SnO_2$ and $G/Sb-SnO_2$ have higher adsorptive capacities than the bare graphene. As the surface area of the graphene and composite materials are all similar, we can conclude that the surface area is not responsible for higher uptake capacity of the composite adsorbents. The difference in the adsorptive capacity can be either due to contribution of the metal oxide in adsorption or the better dispersion of the nanocomposite in the water (i.e., agglomeration of the bare graphene particles may lead to a loss of adsorptive capacity).

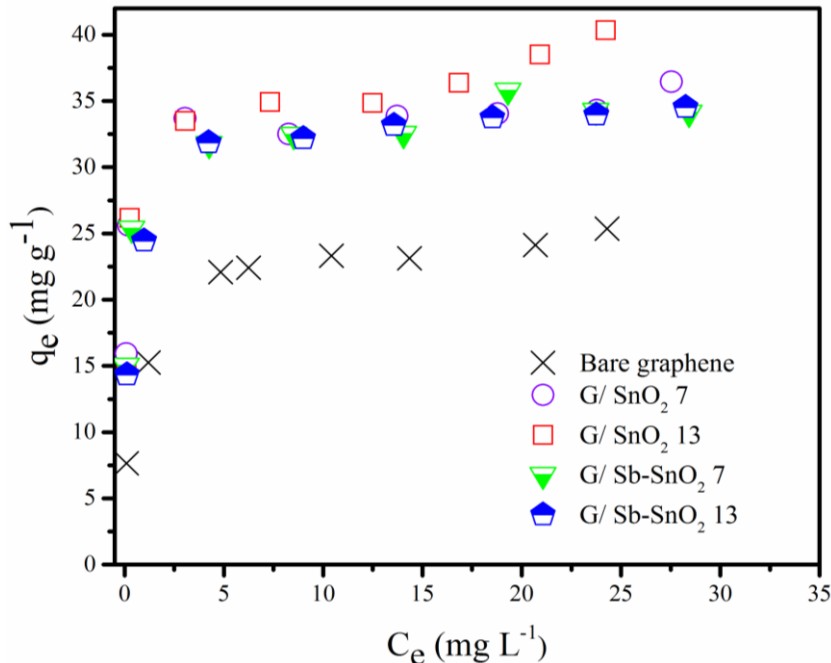

**Figure 4.** Equilibrium adsorption of methylene blue (MB) on bare graphene, G/SnO$_2$ 7, G/SnO$_2$ 13, G/Sb-SnO$_2$ 7, and G/Sb-SnO$_2$ 13.

**Table 1.** Specific surface area of the graphene loaded with metal oxide.

| Sample | Bare Graphene | G/SnO$_2$ 7 | G/SnO$_2$ 13 | G/Sb-SnO$_2$ 7 | G/Sb-SnO$_2$ 13 |
|---|---|---|---|---|---|
| Surface area (BET) m$^2$ g$^{-1}$ | 70 | 84 | 89 | 72 | 69 |
| Surface area (MB) m$^2$ g$^{-1}$ | 62 | 88 | 98 | 85 | 85 |

The Langmuir and Freundlich adsorption isotherm models were fitted to the experimental data for MB adsorption on bare graphene, G/SnO$_2$ 7, G/SnO$_2$ 13, G/Sb-SnO$_2$ 7, and G/Sb-SnO$_2$ 13. The Langmuir and Freundlich isotherms were used to evaluate the equilibrium conditions, and they can be expressed by the mathematical Equations (1) and (2), respectively.

$$\text{Langmuir isotherm} \qquad q_e = \frac{K_l b C_e}{1 + b C_e} \tag{1}$$

$$\text{Freundlich isotherm :} \qquad q_e = K_f C_e{}^n \tag{2}$$

where $q_e$ is the loading of adsorbate on the adsorbent in equilibrium with a solution concentration of $C_e$, $K_L$ and $b$ are the Langmuir isotherm constants, and $K_F$ and $n$ are the Freundlich isotherm constants.

Tables 2–4 show the isotherm constants and the determination coefficient ($R^2$) obtained by nonlinear regression for each adsorbent. By comparing the value of $R^2$ for each adsorbent, it can be observed that Langmuir isotherm provides a better fit to the experimental data than the Freundlich isotherm model. Separation factor ($R_L$) [36] (Equation (3)) and the magnitude of $n$ [37] are important parameters in the Langmuir and Freundlich isotherm as they can indicate whether the adsorption is favorable or not. Area value of $R_L$ or $n$ of less than one indicates favorable adsorption.

$$R_L = \frac{1}{1 + b C_0} \tag{3}$$

**Table 2.** Langmuir constants for adsorption of MB on Bare graphene, G/SnO$_2$ 7, G/SnO$_2$ 13, G/Sb-SnO$_2$ 7, and G/Sb-SnO$_2$ 13.

| | Langmuir Parameters: $q_e = \frac{KbC_e}{1+bC_e}$ | | | | |
|---|---|---|---|---|---|
| Sample | Maximum Adsorption (mg g$^{-1}$) | K | b | R$^2$ | $R_L = \frac{1}{1+bC_0}$ |
| Bare graphene | 25 | 24.3 | 2.28 | 0.92 | 0.017 |
| G/SnO$_2$ 7 | 36 | 34.53 | 11.92 | 0.97 | 0.003 |
| G/SnO$_2$ 13 | 40 | 37.02 | 8.16 | 0.94 | 0.005 |
| G/Sb-SnO$_2$ 7 | 35 | 33.82 | 8.87 | 0.97 | 0.005 |
| G/Sb-SnO$_2$ 13 | 35 | 33.47 | 5.19 | 0.94 | 0.008 |

**Table 3.** Dimensionless Langmuir constants ($R_L$) for adsorption of MB on bare graphene at low and high concentrations for different adsorbents.

| $R_L = \frac{1}{1+bC_0}$ at Two Different Initial Concentrations | | |
|---|---|---|
| | 10 | 50 |
| Bare graphene | 0.04 | 0.009 |
| G/ SnO$_2$ 7 | 0.008 | 0.002 |
| G/ SnO$_2$ 13 | 0.012 | 0.002 |
| G/ Sb-SnO$_2$ 7 | 0.011 | 0.002 |
| G/ Sb-SnO$_2$ 13 | 0.019 | 0.004 |

**Table 4.** Freundlich constants for adsorption of MB on bare graphene, G/SnO$_2$ 7, G/SnO$_2$ 13, G/Sb-SnO$_2$ 7, and G/Sb-SnO$_2$ 13.

| | Freundlich Parameters: $q_e = KC_e^n$ | | | |
|---|---|---|---|---|
| Sample | Maximum Adsorption (mg g$^{-1}$) | K | n | R$^2$ |
| Bare graphene | 25 | 14.90 | 0.17 | 0.92 |
| G/SnO$_2$ 7 | 36 | 26.44 | 0.09 | 0.83 |
| G/ SnO$_2$ 13 | 40 | 26.63 | 0.12 | 0.89 |
| G/Sb-SnO$_2$ 7 | 35 | 25.01 | 0.11 | 0.88 |
| G/ Sb-SnO$_2$ 13 | 35 | 23.39 | 0.13 | 0.91 |

It can be seen from the $R_L$ values in Table 3 that the adsorption process of MB on graphene nanocomposites is favorable at all concentrations, particularly at low concentrations. The values of $n$ obtained for the Freundlich isotherm are consistent with the with $R_L$ values.

The surface area of the nanocomposite can be estimated using MB. As the adsorption of MB on the nanocomposites follows the Langmuir isotherm, it can be concluded that a monolayer of MB has been adsorbed on the surface. It has been reported that one molecule of MB can occupy 130 Å$^2$ on the surface of the adsorbent [38,39]. Table 1 shows the surface area for each adsorbent. It can be seen surface area calculated using BET and MB are more or less the same.

## 2.4. Electrochemical Regeneration

Figure 5a,b and Figure S1a,b demonstrate the evolution of the regeneration efficiency with the charge passed through the electrolytic cell using NaCl and Na$_2$SO$_4$ electrolytes. For all of the adsorbents in both electrolytes, the regeneration efficiency shows a steep increase as the charge was increased from 0 to 1000 C g$^{-1}$, and then it slowly increases until it reaches complete regeneration. The findings reveal that the adsorptive capacity of all of the adsorbents can be completely restored.

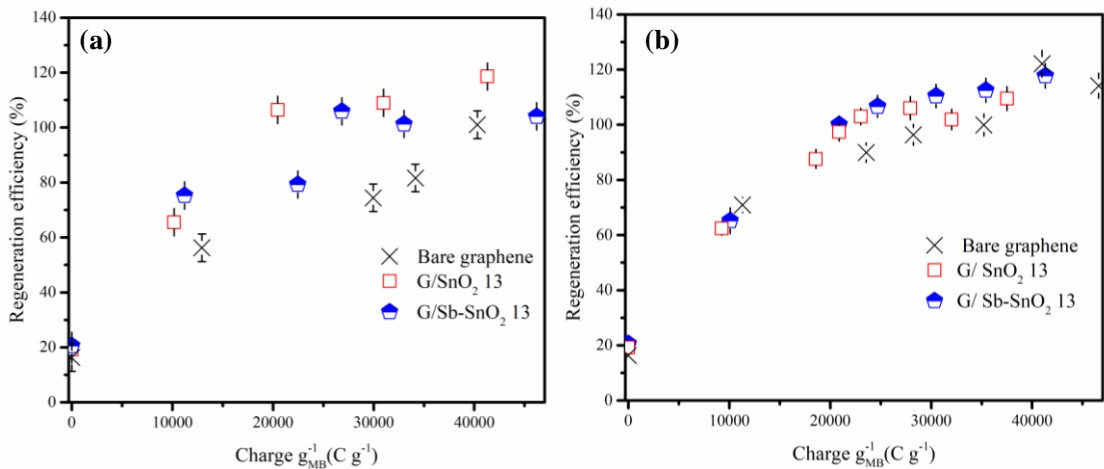

**Figure 5.** Effect of charge passed on regeneration efficiency of MB adsorption on bare graphene, G/SnO$_2$ 13, and G/Sb-SnO$_2$ 13. (**a**) NaCl electrolyte and (**b**) Na$_2$SO$_4$ electrolyte (current density of 10 mA cm$^{-2}$).

In previous studies, it has been reported that the utilization of NaCl instead of Na$_2$SO$_4$ as the electrolyte in the regeneration cell leads to higher regeneration efficiency [8,40]. However, the results of this study suggest that the choice of electrolyte has minimal impact on the regeneration efficiency. Moreover, with sodium sulfate as the electrolyte, if more charge is passed through the cell after complete regeneration of the spent adsorbents, the adsorptive capacity increases less compared to regeneration with a sodium chloride electrolyte. Previous studies have indicated that electrochemical regeneration efficiencies of greater than 100% are associated with oxidation of graphite adsorbent surfaces [12,15]. Thus, the previous studies suggest that with sodium sulfate, less oxidation of the adsorbent's surface occurs, allowing it to remain almost intact. A secondary benefit of using the sodium sulfate electrolyte is that it can hinder the formation of toxic chlorinated hydrocarbons [11].

Previous work has shown that complete regeneration of graphene TiO$_2$ nanocomposite loaded with MB dye could be achieved by passing a charge of 21 C mg$^{-1}$ through the cell [29]. The two main problems associated with this nanocomposite compared to bare graphene were lower adsorptive capacity and a higher cell voltage required for electrochemical regeneration. As explained previously, G/SnO$_2$ and G/Sb-SnO$_2$ showed higher adsorptive capacity compared to bare graphene.

Table 5 shows the specific charge required for complete regeneration of the loaded adsorbents. The prepared nanocomposites had a required charge for complete oxidation as low as 21 C mg$^{-1}$ for G/SnO$_2$ using NaCl as an electrolyte and for G/Sb-SnO$_2$ using Na$_2$SO$_4$ as an electrolyte, which is comparable to the graphene TiO$_2$ nanocomposite results previously reported. Energy consumption for 100% regeneration of the loaded adsorbents is calculated by multiplying the required charge by the cell voltage. As shown in Table 5, the G/SnO$_2$ and G/Sb-SnO$_2$ nanocomposites demonstrate lower charge requirements for complete regeneration than bare graphene using NaCl or Na$_2$SO$_4$ as the electrolyte, with similar cell voltages ($\approx$2.6 V). Thus the G/SnO$_2$ and G/Sb-SnO$_2$ adsorbents require less energy for complete regeneration. These results confirm the higher electrocatalytic activity of the G/SnO$_2$ and G/Sb-SnO$_2$ nanocomposites for regeneration compared to bare graphene.

**Table 5.** Specific charge required for 100% regeneration efficiency of bare graphene, G/SnO$_2$ 7, G/SnO$_2$ 13, G/Sb-SnO$_2$ 7, and G/Sb-SnO$_2$ 13 adsorbents loaded with MB, (regeneration at 10 mA cm$^{-2}$).

| Adsorbent: | Bare Graphene | G/ SnO$_2$ 7 | G/ SnO$_2$ 13 | G/Sb-SnO$_2$ 7 | G/Sb-SnO$_2$ 13 |
|---|---|---|---|---|---|
| Charge passed NaCl (C mg$^{-1}$) | 39 | 21 | 21 | 27 | 27 |
| Charge passed Na$_2$SO$_4$ (C mg$^{-1}$) | 35 | 23 | 23 | 21 | 21 |
| Cell Voltage (V) | $\approx$2.6 | $\approx$2.6 | $\approx$2.6 | $\approx$2.6 | $\approx$2.6 |

## 2.5. Adsorption/Regeneration Cycles

Multiple loading and electrochemical regeneration cycles were carried out to investigate the reusability of the G/SnO$_2$ and G/Sb-SnO$_2$ nanocomposites. The adsorbents were utilized to adsorb MB and regenerated for five successive cycles using either 2 wt. % NaCl or Na$_2$SO$_4$ as electrolytes and 10 mA cm$^{-2}$ current density. For each adsorbent, the required electrochemical treatment time for 100% regeneration was estimated based on the data plotted in Figure 5a,b and Figure S1a,b Table 6 shows the mass of adsorbent recovered after each cycle for the five different adsorbent materials. For the G/SnO$_2$ and G/Sb-SnO$_2$ nanocomposites, only around 10% of the mass of adsorbent was lost over five cycles of adsorption and regeneration. The loss was within 1% to 2% of the loss observed from a control experiment to determine the physical losses associated with the recovery of the adsorbent after filtration. In contrast, with the bare graphene adsorbent, around 30% of the mass was lost over the five cycles of adsorption and regeneration, confirming that significant corrosive oxidation of the graphene was occurring during regeneration.

**Table 6.** Mass of the adsorbent recovered after five consecutive cycles of adsorption and regeneration.

| Sample | G/SnO$_2$ 7 | G/SnO$_2$ 13 | G/Sb-SnO$_2$ 7 | G/Sb-SnO$_2$ 13 | Bare Graphene |
|---|---|---|---|---|---|
| **Mass** | 0.91 | 0.88 | 0.89 | 0.90 | 0.70 |

As shown in Figure 6a,b, the electrochemical oxidation resulted in no reduction of regeneration efficiency, even after five cycles. Figure 6a,b also shows that the change in regeneration efficiency of the bare graphene was much more than the G/SnO$_2$ and G/Sb-SnO$_2$ nanocomposites. This increase in the regeneration efficiency was due to the oxidation of the bare graphene during electrochemical regeneration, which led to the creation of more adsorption sites, either by increasing the surface area or the number of functional groups [15,41]. The adsorptive capacity and consequently regeneration efficiency of the nanocomposites changed less than for bare graphene, which suggests that the addition of the SnO$_2$ and Sb-SnO$_2$ provides some protection of the surface of the graphene from oxidation. By comparing Figure 6a,b, it is evident that the regeneration efficiency of the adsorbents in sodium sulfate increased less than that observed with sodium chloride. This result is consistent with the charge loading results (Figure 5a,b, and Figure S1a,b and confirms that there is a greater tendency for surface oxidation of the graphene using NaCl electrolyte.

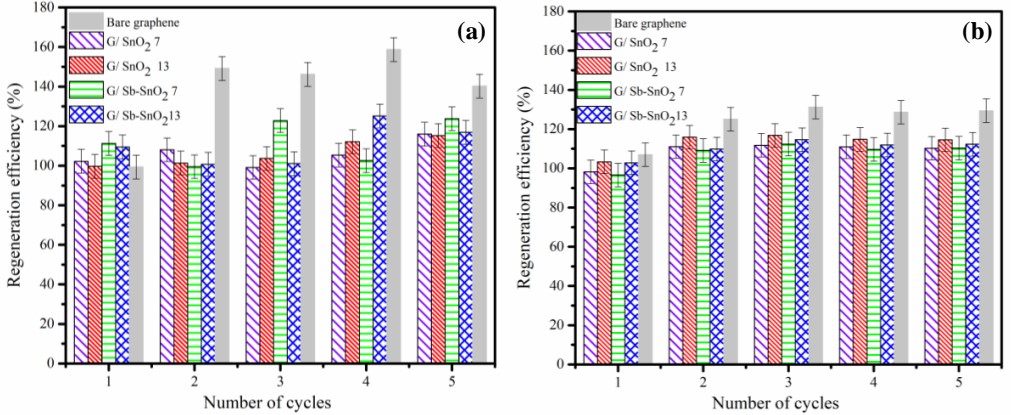

**Figure 6.** Regeneration efficiency for a series of adsorption and electrochemical regeneration cycles of MB adsorption on bare graphene, G/SnO$_2$ 7, G/SnO$_2$ 13, G/Sb-SnO$_2$ 7, and G/Sb-SnO$_2$ 13 nanocomposites in (**a**) NaCl electrolyte and (**b**) Na$_2$SO$_4$ electrolyte. Adsorption was carried out under conditions that give close to the maximum loading of MB on the adsorbent. Regeneration of the bare graphene, G/SnO$_2$ 7, G/SnO$_2$ 13, G/Sb-SnO$_2$ 7, and G/Sb-SnO$_2$ 13 adsorbents were carried out for 14, 10, 10, 12, and 12 min, respectively, at a current density of 10 mA cm$^{-2}$.

A common major problem associated with using NaCl in an electrochemical water treatment process is the formation of chlorinated compounds, which may be more toxic than the primary pollutants [42]. Thus, in addition to reduced oxidation of the graphene, sodium sulfate can be a lower-risk alternative to sodium chloride for regeneration of $G/SnO_2$ and $G/Sb-SnO_2$ nanocomposite adsorbents.

### 2.6. Electrochemical Characterization of the Adsorbents

#### 2.6.1. Linear Sweep Voltammetry

In order to assess the electrochemical characteristics of the nanocomposites, linear sweep voltammetry (LSV) was carried out. Figure 7a shows the results for bare graphene, $G/SnO_2$, and $G/Sb-SnO_2$ nanocomposites, in which the current flow was measured as the applied potential was increased. The experiments were performed in 0.5 M $Na_2SO_4$ at a scan rate of 100 mVs$^{-1}$, with Ag/AgCl and platinum wire used as reference and counter electrodes respectively. Onset potentials of 1.65, 1.87, 1.87, 1.9, and 1.87 V were measured for bare graphene, $G/SnO_2$ 7, $G/SnO_2$ 13, $G/Sb-SnO_2$ 7, and $G/Sb-SnO_2$ 13 adsorbents, respectively, for the oxygen evolution reaction. The delay in the onset potential indicates the suppression of this side reaction and increasing the charge efficiency for generating reactive oxygen species. Such a trend has also been observed by different researchers for $SnO_2$ and $Sb-SnO_2$ electrocatalyst materials [21,25,43,44].

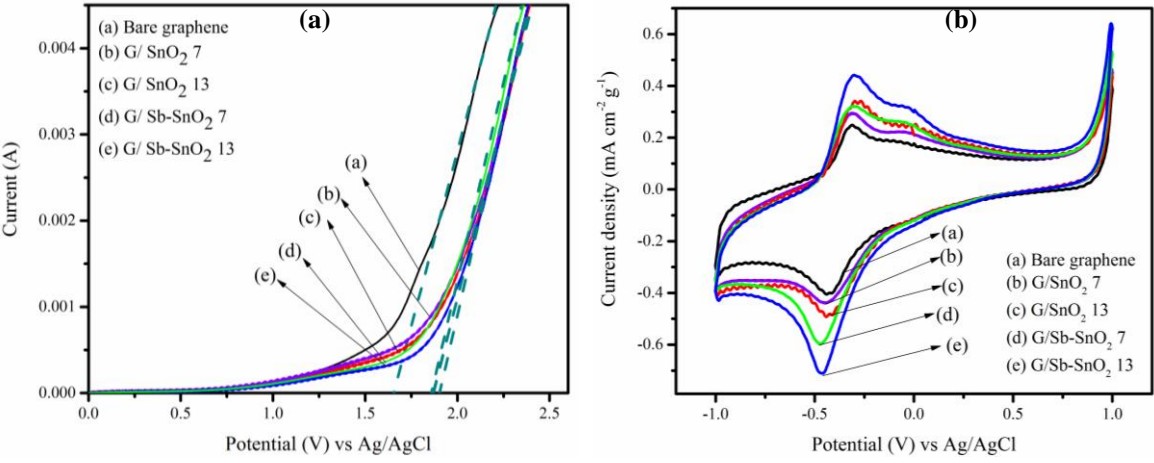

**Figure 7.** (**a**) Linear sweep voltammograms of (a) bare graphene, (b) $G/SnO_2$ 7, (c) $G/SnO_2$ 3, (d) $G/Sb-SnO_2$ 7, and (e) $G/Sb-SnO_2$ 13. Experiments were conducted using 0.5 mol L$^{-1}$ sodium sulfate at a scan rate of 100 mV S$^{-1}$ (**b**) Cyclic voltammetry of (a) bare graphene, (b) $G/SnO_2$ 7, (c) $G/SnO_2$ 13, (d) $G/Sb-SnO_2$ 7, and (e) $G/Sb-SnO_2$ 13 at a scan rate of 10 mV S$^{-1}$ using a solution containing 25 ppm MB and 0.5 mol L$^{-1}$ sodium sulphate.

#### 2.6.2. Cyclic Voltammetry

Figure 7b shows the cyclic voltammogram of the bare graphene, $G/SnO_2$, and $G/Sb-SnO_2$ nanocomposites in 0.5 mol L$^{-1}$ $Na_2SO_4$ solution containing 25 mg L$^{-1}$ of MB. The current density was normalized by the graphene loading for all of the adsorbents. Unlike the bare graphene, $G/SnO_2$ and $G/Sb-SnO_2$ demonstrate a well-defined pair of redox peaks for MB oxidation and reduction. In addition, the peak separation, $\Delta E_p$, was smaller for $G/SnO_2$ and $G/Sb-SnO_2$ nanocomposites relative to bare graphene, demonstrating that the electron transfer was faster for these nanocomposites. It can be seen that the $SnO_2$ and $Sb-SnO_2$ nanoparticles-coated graphene show a higher oxidation peak than that of bare graphene. This implies that the addition of $SnO_2$ and $Sb-SnO_2$ nanoparticles to the graphene increases the electrocatalytic activity, apparently due to synergic effects of the presence of both an n-type semiconductor with a large band gap and graphene [45,46]. It can also be seen that by

increasing the nanoparticles loading from 7 to 13 wt. %, the current density increased for both $G/SnO_2$ and $G/Sb-SnO_2$ nanocomposites. Furthermore, at the same loading of the nanoparticles, $G/Sb-SnO_2$ shows better catalytic activity compared to $G/SnO_2$ nanocomposites, as the $Sb-SnO_2$ acts similar to a metal electrocatalyst and exhibits a high overpotential of oxygen evolution, which can increase its electrocatalytic activity [47–49].

## 3. Experimental

Nanocomposite preparation: a known volume of as-received sol of $SnO_2$ (15 wt. %) or $Sb-SnO_2$ (20 wt. %) (Nyacol Inc., Ashland, MA, USA) was added to 150 mL of a suspension of graphene (GNPs 25 M, XG sciences Inc., Lansing, MI, USA) in DI water (containing 1 g $L^{-1}$ of graphene) and mixed for 24 h. The mixture was dried at 70 °C for 12 h. In this paper, the nanocomposites are named with respect to their composition: thus $G/SnO_2$ 7 corresponds to a graphene/$SnO_2$ composite with a loading of 7 wt. % $SnO_2$. Similarly, $G/SnO_2$ 13 has a loading of 13 wt. % $SnO_2$, and $G/Sb-SnO_2$ 7 and $G/Sb-SnO_2$ 13 have loadings of 7 wt. % and 13 wt. % $Sb-SnO_2$, respectively.

Characterization: The morphologies of the prepared nanocomposites were characterized using scanning electron microscopy (SEM) on a Zeiss Supra55 field-emission SEM (Carl Zeiss Microscopy LLC, White Plains, NY, USA) and transmission electron microscopy (TEM) using a Tecnai TF20 G2 FEG-TEM (FEI, Hillsboro, OR, USA) with a 200−kV acceleration voltage. A Witec alpha 300 R Confocal Raman Microscope (Witec GmbH, Ulm, Germany) was utilized to obtain Raman spectra using a 532 nm laser. The surface area was measured with $N_2$ physisorption (TriStar 3000, Micromeritics Instrument Corp., Norcross, GA, USA) at −196 °C. Before measurement, all samples were degassed at 150 °C for 12 h. The specific surface area was calculated using the Brunauer¬–Emmett–Teller (BET) method in the relative pressure ($P/P_o$) range of 0.01 e 0.99. The concentration of MB in the synthetic wastewater was measured using UV-Visible absorption spectroscopy (UV-2600, Shimadzu, Columbia, MD, USA) at a wavelength of 664 nm [50].

Adsorption study: to obtain the adsorption isotherm, experiments were carried out via a 'bottle point' method. Briefly, 0.1 g of adsorbent was mixed for 30 minutes with 150 mL of the MB solution with different concentrations at room temperature. Synthetic wastewater was prepared using deionized water and MB only. The treated water was filtered, and the filtrate concentration was measured using UV-Visible spectrophotometer.

Electrochemical regeneration: an electrolytic cell was used to regenerate the spent adsorbents using a graphite plate as the anode and a stainless-steel plate as the cathode, as described in an earlier study [14]. Sodium chloride and sodium sulfate (VWR, Radnor, PA, USA) were used as electrolytes. A constant current density of 10 mA $cm^{-2}$ was applied to the cell using a Metrohm Autolab PGSTAT potentiostat (Metrohm AG, Herisaum, Switzerland). The regeneration efficiency was calculated as the ratio of the adsorptive capacity after regeneration to the initial adsorptive capacity, each measured under the same adsorption conditions [51,52].

Electrochemical properties: electrochemical measurements, including cyclic voltammetry (CV) and linear sweep voltammetry (LSV) were performed using the Autolab PGSTAT potentiostat. A volume of 100 mL of a 0.5 mol $L^{-1}$ $Na_2SO_4$ containing 25 ppm MB (VWR, Radnor, PA, USA) was used as an electrolyte. Modified glassy carbon with nanocomposites, Ag/AgCl, and platinum wire were used as working, reference, and counter electrodes, respectively. The modified glassy carbon was prepared by drop-casting a suspension of the nanocomposites (1 mg of adsorbent in 1 mL of Nafion solution, from IonPower, New Castle, DE, USA, with a Nafion-to-adsorbent mass ratio of ~0.1) on a glassy carbon electrode and drying at 70 °C. CV was carried out in a potential range of −1.0 V to +1.0 V at a scanning rate of 10 mV $S^{-1}$.

## 4. Conclusions

This study reported the application of $G/SnO_2$ and $G/Sb-SnO_2$ nanocomposites in an adsorption and electrochemical regeneration process, using both the high conductivity of the graphene and the

large band gap of the tin oxide to improve regeneration efficiency. The adsorptive capacity obtained for the nanocomposites was ≥35 mg g$^{-1}$, which is approximately 1.4- and 1.75-fold higher than the bare graphene and previously reported graphene titanium oxide [29], respectively.

All the prepared nanocomposites showed the ability to attain 100% regeneration efficiency in both NaCl and Na$_2$SO$_4$ electrolytes. With the assistance of SnO$_2$ and Sb-SnO$_2$, the charge efficiency of the process was significantly improved. The results indicate that the required charge passed for complete oxidation of the MB decreased by 50% and 30% for G/SnO$_2$ and G/Sb-SnO$_2$ in an NaCl electrolyte and 35% and 40% for G/SnO$_2$ and G/Sb-SnO$_2$ in an Na$_2$SO$_4$ electrolyte. The reason could be the higher oxygen evolution onset potential along with the higher activity of the SnO$_2$ and Sb-SnO$_2$. Although the cyclic voltammetry results suggested that G/Sb-SnO$_2$ has higher catalytic activity than G/SnO$_2$, the regeneration results show similar charge requirements for complete regeneration for both G/SnO$_2$ and G/Sb-SnO$_2$ nanocomposites. The higher catalytic activity of the Sb-SnO$_2$-modified graphene observed is consistent with previous studies that have demonstrated the suitability of this for electrocatalysis for advanced oxidation combined with a high oxygen overpotential [46,47,50]. An advantage of G/SnO$_2$ and G/Sb-SnO$_2$ over TiO$_2$/graphene nanocomposites is that they have less of an impact on the electrical conductivity of the graphene, minimizing the ohmic losses during regeneration, reducing the energy required.

In addition to the nanocomposites having better performance if Na$_2$SO$_4$ is used as the regeneration electrolyte, using a chloride-free electrolyte reduces the risk of creating toxic breakdown products. This makes sodium sulfate a good electrolyte substitute for sodium chloride.

The possibility of formation of breakdown products that may be released into the treated water was not investigated in this study, and further work is needed to explore this aspect. In addition, the effect of the multiple cycles of regeneration on the structure and surface properties of the nanocomposite adsorbent and the lifetime of the adsorbent should also be investigated.

**Supplementary Materials:** The following are available online at http://www.mdpi.com/2073-4344/10/2/263/s1, Figure S1. Effect of charge passed on regeneration efficiency of MB adsorption on bare graphene, G/SnO$_2$ 7 and G/Sb-SnO$_2$ 7, and (a) NaCl electrolyte and (b) Na$_2$SO$_4$ electrolyte (current density of 10 mA cm$^2$).

**Author Contributions:** E.P.L.R.: conceptualization, supervision, funding acquisition, project administration, writing—review and editing; F.S.: formal analysis, investigation, methodology, writing original draft, writing—review and editing. All authors have read and agreed to the published version of the manuscript.

**Funding:** This research has received financial support from the Natural Sciences and Engineering Research Council of Canada (NSERC 435634-2013 and RGPIN-2018-03725) and the Canada Foundation for Innovation (CFI 32613).

**Conflicts of Interest:** The authors declare no conflict of interest. The funders had no role in the design of the study; in the collection, analyses, or interpretation of data; in the writing of the manuscript, or in the decision to publish the results.

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
