# Peer review of "Electrochemical Oxidation of an Organic Dye Adsorbed on Tin Oxide and Antimony Doped Tin Oxide Graphene Composites"

_catalysts, doi:10.3390/catal10020263_

Round 1
Reviewer 1 Report
Article entitled Electrochemical Oxidation of an Organic Dye Adsorbed on Tin Oxide and Antimony Doped Tin Oxide Graphene Composites, written by Farbod Sharif and Edward P. L. Roberts and submitted to Catalysts journal as a draft no 724662 deals with an important issue of dye decomposition.
The article is interesting and should be considered for publication in Catalysts journal. As English is not my native language, I am not able to assess its correctness. However, while reading I found some statements unclear. Below I enclose the list of my comments.
1) Line 11-14 “We have previously reported….” citation required. However, I think it should not be a part of abstract. It could be mentioned in Introduction part, as a reason why research was conducted, as it is demonstrated in lines 67-74.
2) Line 95: Are synthetic wastewater only MB and water? Do they contain any other substances?
3) Line 242 vs 250-251: Have changes in the structure of the tested material/composite been tested?
Based on my comments and general impression I suggest accepting this articles, with small supplementation
Author Response
We are grateful for the reviewers positive comments and feedback. Our response to the detailed comments are provided below.
1) Line 11-14 “We have previously reported….” citation required. However, I think it should not be a part of abstract. It could be mentioned in Introduction part, as a reason why research was conducted, as it is demonstrated in lines 67-74.
We have removed this sentence from the abstract.
2) Line 95: Are synthetic wastewater only MB and water? Do they contain any other substances?
The synthetic wastewater only contained MB and water. A sentence has been added to confirm this in the experimental section.
3) Line 242 vs 250-251: Have changes in the structure of the tested material/composite been tested?
We have not characterized the adsorbents after regeneration in this study.
Reviewer 2 Report
These authors are very well known by the scientific community working on this topic. They have proved to do valuable works for this research area and this work is another example of their great labour. I have only some minor comments about this paper:
Figure 5.b seems a little bit messy, but I understand that the scales are difficult to adjust. Could you please look for the way to make this Figure clearer? Figure 6. Please do not use black colour for this kind of diagrams. It does not attract the reader. I would like to see some future work ideas in the conclusion section. I think that it could give some encouraging paths for the people who follow your work.Thanks a lot for your contribution.
Best regards
Author Response
We are grateful for the reviewers positive comments and feedback. Our response to the detailed comments are provided below.
Figure 5.b seems a little bit messy, but I understand that the scales are difficult to adjust. Could you please look for the way to make this Figure clearer?
The figure has been modified to make it clearer, and some of the data moved to supporting information.
Figure 6. Please do not use black colour for this kind of diagrams. It does not attract the reader.
The black bar has been changed to grey.
I would like to see some future work ideas in the conclusion section. I think that it could give some encouraging paths for the people who follow your work.
A couple of suggestions for future work have been added at the end of the conclusions section.
Reviewer 3 Report
This draft reported an approach to increase regeneration efficiency by synthesizing composites materials including Sn and Sb oxide. They did a through study to support their idea, and I agreed to author’s point. I recommend a minor revision of this manuscript. The below is my comments on this work.
Page 1 ling 32, a ‘comma’ is misplaced in the sentence. Pape 2 ling 71, they said that calcination led to low adsorptive capacity of adsorbents. Why calcination degrades the performance? Need more explanation. Page 4 line 135, I think surface energy of nanoparticle resulted in the aggregation, not Van der walls force. Page 5 line 155, Why better dispersion of the nanocomposite in water increased adsorption capacity? Explain more. Based on your data, G/Sb-SnO2 was the best in your sample. What make you select Sb-Sn combination as the possible adsorbent? I understood choice of Sn, but not addition of Sb. I found typos and errors throughout the manuscript although I didn’t pay attention to check them. Please revise any mistakes made in the draft.
Author Response
We are grateful for the reviewers positive comments and feedback. Our response to the detailed comments are provided below.
Page 1 ling 32, a ‘comma’ is misplaced in the sentence.
The sentence has been revised.
Pape 2 ling 71, they said that calcination led to low adsorptive capacity of adsorbents. Why calcination degrades the performance? Need more explanation.
Calcination used in the preparation of the nanocomposites leads to loss of amorphous carbon. An explanation has been added to the manuscript, and readers can also refer to the cited reference for further details.
Page 4 line 135, I think surface energy of nanoparticle resulted in the aggregation, not Van der walls force.
This has been changed to surface energy.
Page 5 line 155, Why better dispersion of the nanocomposite in water increased adsorption capacity? Explain more.
Agglomeration of the bare graphene may lead to a loss of available surface area and hence adsorption capacity. This explanation has been added in the revised manuscript.
Based on your data, G/Sb-SnO2 was the best in your sample. What make you select Sb-Sn combination as the possible adsorbent? I understood choice of Sn, but not addition of Sb.
Many studies in the literature have demonstrated that antimony doped tin oxide is an excellent electrocatalyst for advanced oxidation combined with a high oxygen overpotential. Previous work reporting the suitability of Sb-SnO2 are cited in the introduction (3rd Paragraph). In fact our results show similar perofrmance for the SnO2 and Sb-SnO2 nanocomposites, although the electrochemical characterisation does indicate enhanced electrocatalysis. A sentence has been added to the conclusions, commenting that the observed activity of the Sb-SnO2 nanocomposite is consistent with previous studies, citing relevant references.
I found typos and errors throughout the manuscript although I didn’t pay attention to check them. Please revise any mistakes made in the draft.
We have been through the manuscript to correct any typos and errors we can find.